# Alleviation of Sodic Stress in Rice by Exploring the Exopolysaccharide-Producing Sodic-Tolerant Bacteria

Yazhini Gunasekaran [1], Subramaniam Thiyageshwari [1,*], Manikandan Ariyan [2], Aritra Roy Choudhury [3], Jung-Ho Park [3], Duraisamy Selvi [1], Lakshmanan Chithra [4] and Rangasamy Anandham [2,*]

[1] Department of Soil Science and Agricultural Chemistry, Tamil Nadu Agricultural University, Coimbatore 641003, Tamil Nadu, India
[2] Department of Agricultural Microbiology, Tamil Nadu Agricultural University, Coimbatore 641003, Tamil Nadu, India
[3] Bio-Evaluation Center, Korea Research Institute of Bioscience and Biotechnology, Cheongju 28116, Korea
[4] Department of Soil Science and Agricultural Chemistry, Horticultural College and Research Institute for Women, Tiruchirappalli 620027, Tamil Nadu, India
\* Correspondence: thiyageshwari@gmail.com (S.T.); anandhamranga@gmail.com (R.A.)

**Abstract:** Sodicity is one of the major salt stresses that impair crop production. Exopolysaccharide-producing sodic tolerant bacteria (EPS-STB) play a significant role in reducing the sodic stress in plants by hampering the uptake of sodium. In this context, this study aims to isolate the EPS-STB for alleviating sodic stress in rice under a sodic environment. Thus, artificial sodicity was created in culture media, and 253 bacteria were isolated from the rice rhizosphere of sodic soils in Trichy and Chinna Salem of Tamil Nadu in India. Fifty bacterial isolates were initially screened based on EPS production, sodic tolerant ability, and plant growth-promoting activities. Further, these bacterial isolates were identified using 16S rDNA sequencing. The results suggested that the isolated bacteria possessed biofilm-forming abilities along with plant growth-promoting activities and osmolyte accumulation under sodic stress conditions. *Bacillus rugosus* L1C7T, *Bacillus paralicheniformis* L1C5L, *Pseudomonas* sp. L5C14T and *Franconibacter helveticus* L2C1L2 were chosen as better EPS-STB plant growth-promoting bacteria, and their impact on rice under sodic conditions was evaluated. Among the sodic tolerant bacteria, *Franconibacter helveticus* L2C1L2-inoculated rice plants increased dry matter production compared to the control. Thus, this study showed that the utilization of EPS-STB will become a promising tool to alleviate sodic stress in rice.

**Keywords:** exopolysaccharide; sodic-tolerant bacteria; plant growth-promoting traits; biofilm formation; rice-sodic soil

## 1. Introduction

The salt-affected soils are classified into saline, sodic and saline-sodic soils, and each of them affects the crop with a distinct mechanism. Among these, sodicity is one of the greatest obstacles to crop production. Worldwide, 618 million ha of land is affected by sodicity [1], and in India, it has been estimated around 3.77 million ha [2]. Plant growth impairment occurs under salt accumulation in two phases: i) osmotic stress induces water and nutrient deficit in the plant by affecting water and nutrient uptake; ii) ionic toxicity ($Na^+$) accumulates in the cytosol [3]. High sodium ion accumulation in plants reduces photosynthetic activities and causes cell death, membrane injury, DNA damage, and degradation of proteins by the production of reactive oxygen species (ROS) [4,5]. As a result, systemic tolerance of plants to sodic stress is required to attain sustainability in crop production under stress conditions. Rhizobacteria can colonise plant roots and foster plant growth both directly and indirectly [6]. Previous research has shown that plant growth-promoting rhizobacteria (PGPR) could produce plant growth-promoting hormones such as indole acetic acid, gibberellic acids, cytokinin, solubilization of unavailable nutrient forms,

nitrogen fixation, and synthesis of ACC deaminase to boost plants to withstand stress [7]. Inoculation of PGPR was advocated to increase the growth of rice, wheat, and tomato under sodic stress conditions [8,9]. Previous studies have shown that the discovered bacteria from several genera, including *Agromyces*, *Pseudomonas*, *Bacillus*, *Paenibacillus*, *Enterobacter*, and *Burkholderia*, displayed features that promoted plant growth in their host plants when exposed to sodicity. Some PGPRs have been found to indirectly improve plant development by acting as pathogen-fighting antimicrobials [10,11].

Exopolysaccharides (EPS) are complex polymers composed mainly of polysaccharides, proteins, nucleic acids, and humic acids [8]. The complex layer produced around the microorganism can shield them against adverse conditions and is responsible for the interaction between the microorganism and the adhesion of biofilm to the surface [12]. Moreover, soil aggregates are stabilized by these polymers by a variety of interactions, including dispersion forces, electrostatic interactions, and hydrogen bonds [13]. Therefore, EPS-producing PGPR could increase the macropore volume and rhizosphere soil aggregation, resulting in more water and nutrient availability to the crop [14]. Inoculating the plant with EPS-producing bacteria has been shown to reduce the plant's uptake of sodium ions from the soil. Therefore, enhancing EPS-producing sodic-tolerant plant growth-promoting bacteria colonisation is necessary to hamper sodium uptake, thereby alleviating the sodic stress in plants.

Numerous studies have investigated the connection between EPS and plant growth enhancement in salinity and saline-sodic environments [5,7,15–17]. There is a dearth of information on the protection provided by EPS against sodicity, and the majority of the cited literature used sodium chloride to test bacteria sodic tolerance, even though NaCl is a neutral salt that does not raise pH. Not only is the concentration of sodium ions (NaCl) an important factor in screening for sodic-tolerant strains, but elevated pH is also the most important factor in determining the bacterial population of the sodic environment. Therefore, the purpose of this study is to isolate and characterise STB that produces EPS from the rhizosphere of rice cultivated in sodic soils and to alleviate sodicity in rice under in vitro conditions.

## 2. Materials and Methods

### 2.1. Bacteria Isolation

The rice plants were collected from five different sodic soil in Trichy and Chinna Salem, both in Tamil Nadu, India, having pH 8.5–9.2, EC-0.2–1.2 dSm$^{-1}$. The exchangeable sodium percentage (ESP) in Trichy was 15 to 16, and in Chinna Salem, the ESP was 19 and 24 (Table S1). The range of carbonate and bicarbonate content in soil was 0.85 to 2.69 g kg$^{-1}$ and 0.01 to 0.02 g kg$^{-1}$, respectively. The rhizosphere soil of rice crops was collected at the vegetative stage (45-day-old seedlings). A quadratic method was employed to collect the rice plant, with three replications in each place. Four plants were collected in each quadrant. As a result, a total of twelve plants were collected from each location. The collected rice roots were vigorously shaken to remove loosely adhered soil, and the tightly adhered soil was collected separately in a sterile polythene bag and stored at 4 °C. Then, the samples from each quadrant were combined into a single sample. To isolate the rhizosphere bacteria from rice-sodic soil, ten grams of rhizosphere soil was added to 100 mL of sterile distilled water in a 250 mL conical flask and shaken for 20 min at 250 rpm. Then the homogenized aliquot was serially diluted up to $10^{-7}$, and 100 μL of an aliquot from $10^{-5}$, $10^{-6}$, and $10^{-7}$ were spread on the agar plates. To obtain more bacterial colonies from the sodic soil, nutrient-rich media (Nutrient agar, Luria Bertani agar, R2A, and Tryptic soy broth) (HiMedia, Mumbai, India), as well as low-nutrient media (1/10 NA, 1/10 LB, 1/10 R2A, and 1/10 TSA), were used. The plates were incubated for 48 h at 30 °C and surveyed every 24 h for new bacterial colonies. The colonies were differentiated based on their size, colour, morphology, and 253 colonies were selected and transferred to new plates. After five to six successive streakings, the culture purity was ascertained by examination under a light microscope. The purified bacterial isolates were maintained in 60% glycerol at −80 °C.

### 2.2. Standardization of Medium for Inducing Sodicity

To induce the sodicity (pH > 8.5, EC < 4 dS m$^{-1}$, and ESP < 15) in a liquid medium, the actual composition of tryptic soy broth (TSB) was slightly modified. Instead of NaCl, 0.5% of $Na_2CO_3$ was added to the TSB, and the pH was adjusted to 8.7 using 0.1 N HCl (after sterilisation in an autoclave, the medium pH was increased to 9.5).

### 2.3. Screening for Exopolysaccharide Production

EPS production of bacterial isolates was quantitatively estimated. Briefly, 150 µL ($10^7$ CFU/mL) of freshly grown overnight culture was inoculated into 15 mL of TSB and incubated for 48 h at 100 rpm at 30 °C. The EPS were separated from the cell by centrifuging 1.5 mL of liquid culture at 10,000 rpm for 15 min at 4 °C. To precipitate polysaccharides, thrice the volume of prechilled acetone was added, mixed thoroughly, and incubated at $-20$ °C for 24 h. The precipitate was harvested by centrifugation at $14,000 \times g$ rpm for 20 min and air-dried to remove the excess acetone. The number of EPS was estimated by the phenol-sulphuric method [18]. To quantify EPS, the precipitate was dissolved in 1 mL of distilled water, and 200 µL of aliquot was taken in a test tube with 800 µL of distilled water. A total of 5 mL of 2% phenol regent followed by 96% sulphuric acid were added, incubated for 30 min for colour development, and measured at 490 nm with a spectrophotometer (UV-spectrophotometer, UV-1601, Shimadzu, Japan). The standard was glucose, and a blank was run simultaneously. Similarly, the EPS production was quantified under sodic conditions using the modified TSB medium as described above.

### 2.4. DNA Extraction

Genomic DNA was extracted from 24 h old culture of bacterial isolates by adopting the standard CTAB method. Gel electrophoresis was performed with 0.8% agarose and ethidium bromide (10 mg mL$^{-1}$) to ascertain the presence of DNA and documented on a gel documentation system (Vilber, Germany). The concentration of DNA of each bacterial isolate was quantified by a Nanodrop spectrophotometer (Thermo Scientific, Nanodrop 2000c, Wilmington, DE, USA). Subsequently, 16S rDNA extracted from isolates was amplified by PCR using the universal primer 27F (5′-AGA GTT TGA TCA TGG CTC AG-3′) and 1492R (5′-GGT TAC CTT GTT ACG ACT T-3′) [19]. The amplification conditions were the initial DNA denaturation at 94 °C for 5 min followed by 40 cycles at 94 °C for 40 s for denaturation, 50 °C for 40 s for annealing, and 72 °C for 90 s for extension, and a final polymerization period of 7 min at 72 °C. Then, 16S rRNA nucleotide sequences were identified by PCR-direct sequencing using the fluorescent dye terminator method (ABI prismTM BigdyeTM Terminator cycle sequencing ready reaction kit V.3.1), and the products were purified with a Millipore–Montage dye removal kit. Finally, the products were run in an ABI 3730XL capillary DNA sequencer (50 cm capillary). The obtained sequences were aligned and blasted in EzTaxon (http://www.ezbiocloud.net/) (accessed on 21 December 2021) to ascertain their identity. The nucleotide sequences were submitted to the NCBI database, and their accession numbers are available in the GenBank OK427230, OM392062-OM392064, OM421749, OM421787-OM421792, OM422610-OM422640, OM486948, OM584324, OM604753, OM614587, OM615901, OM614598, OM615903, OM616571, and OM640465.

### 2.5. Screening of Sodic Tolerance

The intrinsic sodic tolerance ability of isolated bacteria was assessed by inoculating the bacterial culture ($10^7$ CFU/mL) into TSB (pH 7) and modified TSB (pH 9.5) incubated for 24 h at 30 °C at 100 rpm. Then, bacterial growth was determined through a spectrophotometer at 600 nm, and uninoculated liquid media (TSB and modified TSB) was used as a blank.

### 2.6. Screening for Plant Growth Promoting (PGP) Traits

The PGP traits of EPS-producing STB were evaluated under both control and sodic conditions. Gordon and Weber's [20] method was used to examine the indole acetic acid production (IAA) in TSB and modified TSB supplemented with 100μL of 0.1% of L-tryptophan. Briefly, the bacterial isolates ($10^7$ CFU/mL) were inoculated in the above media and incubated for 48 h in the dark. A part of the culture (5 mL) was centrifuged at 8000 rpm for 15 min, and 2 mL of supernatant was transferred to a new tube, which was then added with 100 μL of 10 mM orthophosphoric acid and 4 mL of Salkowasky reagent (1 mL of 0.5 M $FeCl_3$ in 50 mL of 35% $HClO_4$) (HiMedia, Mumbai, India). After 25 min of incubation at room temperature, the absorbance of the pink colour developed was measured at 530 nm (UV-spectrophotometer, UV-1601, Shimadzu, Japan). A standard curve was constructed using pure IAA as a standard to determine the IAA concentration in culture. The ability of isolates to solubilize tricalcium phosphate was tested in Pikovskaya's agar medium supplemented with 0.5% tricalcium phosphate [21]. Similarly, the zinc and silica solubilization abilities of the isolates were evaluated in Bunt and Rovira medium supplemented with 0.1% of ZnO and $MgSiO_2$, respectively. 10 μL ($10^7$ CFU/mL) of each bacterial isolate was spotted in triplicate and incubated for 4 days at 30 °C. Then, the halo zone formed around the bacterial colonies in plates indicated the solubilization of insoluble nutrients (P, Zn, and Si). The solubilization percentage for Zn, P and Si was calculated by following the formula [22].

$$\text{Percentage Solubilization Index} = \frac{\text{colony diameter } + \text{ halozone diameter}}{\text{colony diameter}} \times 100$$

To test the siderophore production, the overnight culture was inoculated in the chrome azurol S agar plate and kept for 4 days at 30 °C, and the production of the orange halo zone around the colony was a positive indicator for siderophore production. Additionally, sulphur oxidation potential was examined using the mineral salts thiosulphate (MST) medium as described previously [23].

### 2.7. Estimation of Osmolytes Production

The bacterial isolate was grown for 24 h at 30 °C in TSB (pH 7.0) and modified TSB (pH 9.5) under shaking conditions (120 rpm). The cells were harvested by centrifugation at $12,000\times g$ rpm for 10 min. The supernatant was used for glycine betaine estimation. The supernatant was diluted 1:1 with 2 N sulphuric acid, an aliquot of 0.5 mL was taken in the test tube, and 0.2 mL of cold iodine was added and stored at 4 °C for 16 h. Then, it was centrifuged at $10,000\times g$ rpm for 15 min at 0 °C, and the precipitate was dissolved in 9 mL of 1,2 dichloroethane. The colour intensity was measured at 365 nm in a spectrophotometer. Glycine betaine was used for standard preparation [24]. In the cell pellet, 80% ethanol was added and incubated at 60 °C for 45 min in the water bath. Then centrifugation was done at 12,000 rpm for 15 min. Trehalose content was estimated from the cell extract by following the phenol-sulphuric acid method [18], and the same cell extract was utilized for proline estimation [25]. Briefly, 1 mL of cell extract acid ninhydrin and glacial acetic acid were added into a test tube and incubated for 1 h at 100 °C in a water bath, then cooled. A total of 2 mL of toluene was added to extract the proline from the reaction mixture. The appearance of pinkish-red colour was measured at 520 nm and working standards were prepared using L-proline.

### 2.8. Biofilm Formation

The biofilm formation ability of the bacteria was quantitatively estimated according to the protocol [26]. Briefly, selected exopolysaccharide-producing salt-tolerant bacteria were grown in a tryptic soy liquid medium at 30 °C at 120 rpm. A total of 10 μL of each bacterial culture and 150 μL of TSB (pH 7) and modified TSB (pH 9.5) were inoculated in an individual well of a 96-well microtitre plate. The plate was covered with a lid and by a thin film and kept in undisturbed condition for 96 h at 30 °C. Then the planktonic growth

was measured at 660 nm, loosely bounded cells on the well were removed, and 0.1% of 250 μL crystal violet was added into each well and allowed for 15 min. The stained wells were washed with distilled water thrice to remove excess indicators, and the titre plate was dried overnight. A total of 250 μL of 30% acetic acid was added into each well to destain the adsorbed crystal violet, and the intensity of colour development was measured as a biofilm-forming ability at 550 nm (Spectra Max i3X, Molecular devices LLC., San Jose, CA, USA). The ratio between A550 nm and A660 nm was used to quantify the ratio of biofilm formation to the planktonic population.

### 2.9. Alleviation of Sodic Stress in Rice

The selected bacteria were examined for their ability to alleviate sodic stress in rice cv. CO-51. To ascertain the rice susceptibility to sodicity, it was grown at various sodium concentrations (0, 0.2 M, 0.4 M, 0.6 M, 0.8 M, 1.0 M, 1.2 M, 1.4 M, 1.6 M, 1.8 M, and 2.0 M). Sodium carbonate was used to achieve the various sodium concentrations, and a pH of 9.5 was maintained at each concentration. Briefly, seeds were immersed in different sodium concentrations overnight before being transferred to Petri dishes with germination sheets. Using the various sodium concentration solutions, the moisture was correctly maintained for a week. The susceptibility to sodium was then calculated based on the growth. The roll towel experiment was conducted to assess the ability of the bacterium to alleviate sodic stress. The viable seeds were surface disinfected with 1% sodium hypochlorite for 1 min, followed by 70% ethanol for 30 sec, and at the end of each step, the seeds were rinsed with sterile water. Then, the disinfected seeds were imbibed with selected bacterial isolates ($10^7$ CFU/mL) overnight. The treatment details were as follows: T1—control, T2—*Bacillus paralicheniformis* L1C5L, T3—*Pseudomonas* sp. L5C14T, T4—*Bacillus rugosus* L1C7T, and T5—*Franconibacter helveticus* L2C1L2 with five replications. Then, the seeds were placed on germination paper (30 cm × 40 cm), rolled, and kept in modified Hoagland solution where 1.0 M of sodium and 200 mM of Ca, and 200 mM of Mg were added, and the pH was adjusted to 9.5 to induce the sodic stress. The solution was changed every week, and the plants were retained for 16 days. The plants were carefully removed from the germination sheet, and height and dry weight were measured. The dried plant sample was ground in a Willey mill and passed through a 20 mesh sieve. Triple acid (Nitric: Sulphuric: Perchloric as 9:2:1) was added to powdered plant samples, and the funnel mouth was covered with a flask and kept over the sand bath until a clear solution was obtained. The digested content was filtered, and the sodium content in the filtrate was analyzed using a flame photometer (Systonic Microprocessor Flame photometer 935, Haryana, India).

### 2.10. Bioassay for Biofilm Forming Ability

The following approach was used to visualise the biofilm-forming ability of bacterial isolate [27]. The rice seeds cv., CO-51 were surface disinfected as described previously and treated with 2% ketoconazole, washed thrice using sterile distilled water, and kept in the dark for 24 h for germination. The uniformly germinated seeds were chosen and carefully placed in a glass tube (30 cm × 2.5 cm diameter) containing 50× Murashige and Skoog medium (MS medium) with 0.5% Phytagel™ (Sigma-Aldrich, St. Louis, MI, USA). Then, the tubes were kept in the plant growth chamber with 12 h light (200 moles m$^{-2}$ s$^{-1}$) at 28 °C and maintained for 20 days. Simultaneously, the bacterial culture was grown in a liquid tryptic soy medium for 24 h. The cells were harvested by centrifugation at 6000× *g* rpm for 15 min. The cell pellets were washed twice with sterile phosphate buffer and resuspended with phosphate buffer ($10^7$ CFU/mL)). A total of 20 μL of culture suspension was inoculated at the hypocotyl region of 5-day-old seedlings, whereas control was added with only tryptic soy medium. The biofilm that formed around the roots was visualised and photographed.

*2.11. Assessing the Intracellular Na$^+$ Concentration in Bacterial Isolates*

Intracellular sodium uptake was assessed with the following procedure [9]. The bacterial isolates were cultured in TSB amended with 0.5% sodium carbonate and incubated for 72 h at 30 °C. Cells were separated by centrifugation at $7000\times g$ rpm for 15 min, the supernatant was discarded, and the cells were washed twice with phosphate buffer to remove the excess medium. The purified cells were digested with nitric acid, and the extract was filtered and diluted to 100 mL. The sodium concentration was estimated using a flame photometer (Systonic Microprocessor Flame photometer 935, Haryana, India).

*2.12. Aggregation Formation*

The potential of EPS on soil aggregate formation was examined in sodic soil. A total of 50 g of sodic soil was taken in 26 plastic cups (200 mL), and their maximum water holding capacity was measured by adopting the standard protocol. The three-day-old bacterial culture was centrifuged, and the collected supernatant was added to 13 plastic cups. In the remaining 13 cups, water was added until up to 60% of the pores were filled and kept for 20 days. Throughout the incubation period, the moisture level in cups was kept constant on a weight basis. The cup was carefully turned upside down to collect soil in an undisturbed manner after 30 days of incubation. In a clean Petri dish (Borosil), a piece of aggregate from each cup and an equal amount of water was added to imbibe it fully and kept for 2 h incubation. Then, the image of soil aggregate dispersity was captured.

*2.13. SEM Imaging of Unstressed and Stressed Cultures*

The efficient sodic tolerance bacterium was grown in a medium amended with and without 0.5% sodium carbonate and kept for incubation at 30 °C for 24 h. The cells were harvested by centrifugation at $6000\times g$ rpm for 15 min and fixed by 2.5% (*v/v*) of glutaraldehyde for 1 h, and 1% (*v/v*) osmium tetroxide was used for post-fixation for 40 min at 4 °C. The dehydration of the specimen was done by using 10% ethanol and visualised through SEM (Quanta 250, FEI, Eindhoven, The Netherland).

*2.14. Statistical Analysis*

The data on sodic tolerance, exopolysaccharide production, and biofilm formation, Principal component analysis (PCA) was carried out to select the efficient bacterial culture using Microsoft Excel for Windows 2010 add-in with XLSTAT version 2022.1.1.1251 (Addinsoft Inc., New York, NY, USA). To group the bacteria based on plant growth-promoting traits and osmolyte accumulations, Tukey's test was performed at a 5% significance level using IBM SPSS Statistics version 22 (New York, NY, USA).

## 3. Results

*3.1. Isolation and Screening of EPS-Producing Bacteria*

From the rice rhizosphere under five different sodic soils and eight different media compositions, 253 bacteria were isolated. The initial screening on EPS production was assessed for 253 isolates. The distribution of bacterial isolates based on their EPS production is depicted in a histogram (Figure 1). The highest concentration of EPS production by the isolated bacteria was 1390 mg/L, and the average EPS production was 355 mg/L. Only a few bacterial isolates produced EPS above 1000 mg/L. The majority of isolates produced EPS in the range of 280–560 mg/L. Among the 253 cultures, based on their EPS-producing capability, the first 50 isolates of higher EPS producers were selected. The screening process for selecting the bacterial isolates is illustrated in Figure S1.

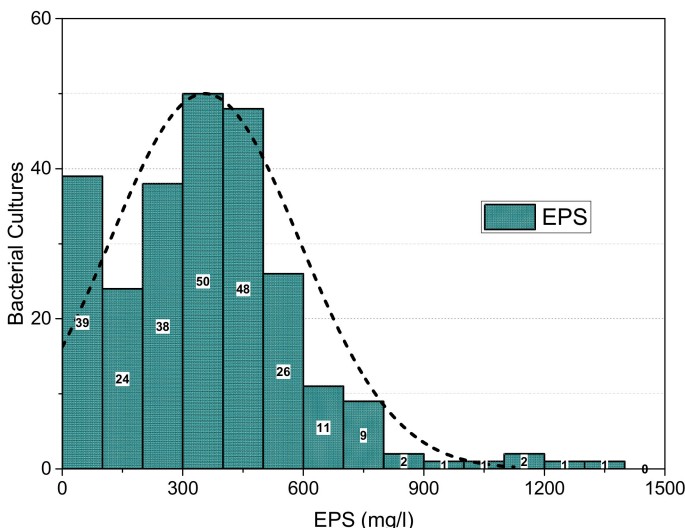

**Figure 1.** EPS-producing capability of isolated bacteria isolated from rice sodic soils.

### 3.2. PCR and 16S rRNA Analysis of Bacterial Isolates

Fifty bacterial isolates with different morphologies and significant EPS production were molecularly characterised. These isolates were identified using 16S rRNA sequencing viz., *Bacillus, Pseudomonas, Stenotrophomonas, Burkholderia, Rhodococcus, Lysinibacillus, Cytobacillus, Ralstonia, Arthrobacter, Niallia, Achromobacter, Staphylococcus, Kocuria, Franconibacter, Planococcus,* and *Acinetobacter* (Table S2). *Bacillus* spp. is the most predominant genus found in these two sodic soils.

### 3.3. Standardization of Media for Inducing Sodicity

A standardization experiment was carried out to choose the best sodium sources for generating sodicity (Table S3). To raise the pH in medium, $Na_2CO_3$-$NaHCO_3$ buffer, $Na_2CO_3$, NaOH, NaCl, and $NaHCO_3$ were used. Before and after the autoclave, the pH was tested. The pH was not raised over 8.5 with NaCl or $NaHCO_3$. Due to the increased dissociation of $Na_2CO_3$-$NaHCO_3$ buffer at a higher temperature during autoclave, the increase in pH could not be controlled when it was added to the liquid medium. After autoclaving, the pH was continually raised by $Na_2CO_3$ and NaOH. Furthermore, sodium is found in the form of sodium carbonate in sodic soil; $Na_2CO_3$ was chosen over NaOH to induce sodicity in a liquid medium mimicking sodic soil.

### 3.4. Screening for Exopolysaccharide Production

The EPS-producing potential under stress conditions was examined with 0.5% $Na_2CO_3$ at pH 9.5 (EPS_S1). The sodicity showed a significant impact on EPS production, which was reduced by six-fold when compared to the control (EPS_C) (Figure 2). The median of EPS production without stress conditions was higher than the sodic stress. The distribution of EPS production in control was slightly skewed to the positive side. However, the distribution under stressed conditions was narrowed, and most of the isolate's EPS production was concentrated within a range.

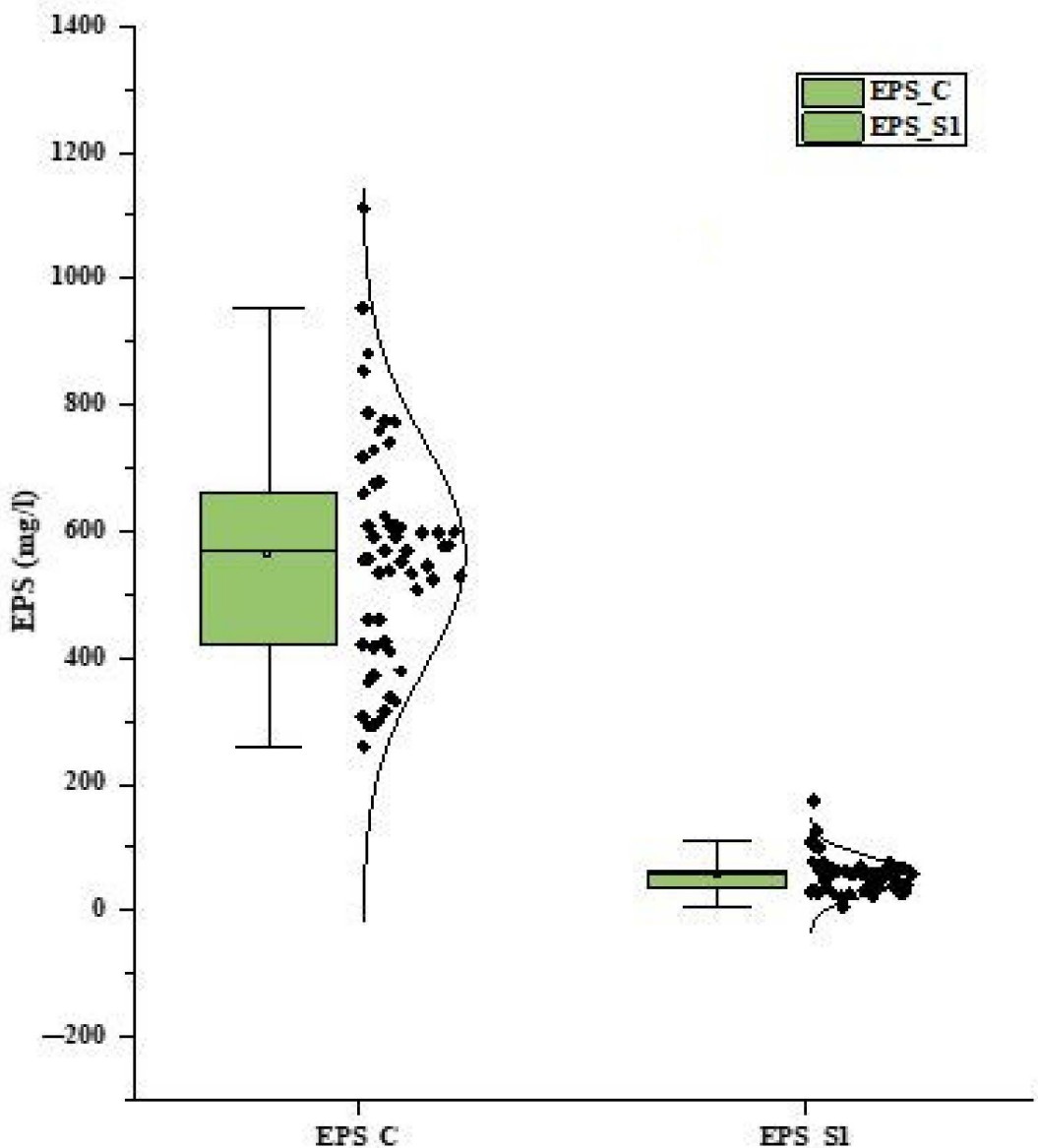

EPS_C- EPS production @ Control, EPS_S1- EPS production @ 0.5 % Na$_2$CO$_3$

**Figure 2.** Distribution of bacterial EPS production at stressed and unstressed conditions. EPS_C—EPS production for control; EPS_S1—EPS production at 0.5% Na$_2$CO$_3$.

*3.5. Screening for Sodicity Tolerance*

The sodicity tolerance of EPS-producing bacterial isolates was checked at two different concentrations, viz., ST_S1 (0.5% Na$_2$CO$_3$) and ST_S2 (0.75% Na$_2$CO$_3$) with control (ST_C). The sodium content in ST_S1 was around 2140 ppm, merely enough to mimic a sodic effect in the growth medium. The distribution of the sodic tolerance ability of the bacteria is shown in Figure 3. The mean optical density of bacterial growth in unstressed conditions was 0.7, which was reduced to half (0.4) under ST_S1 and further reduced to 0.02 under ST_S2.

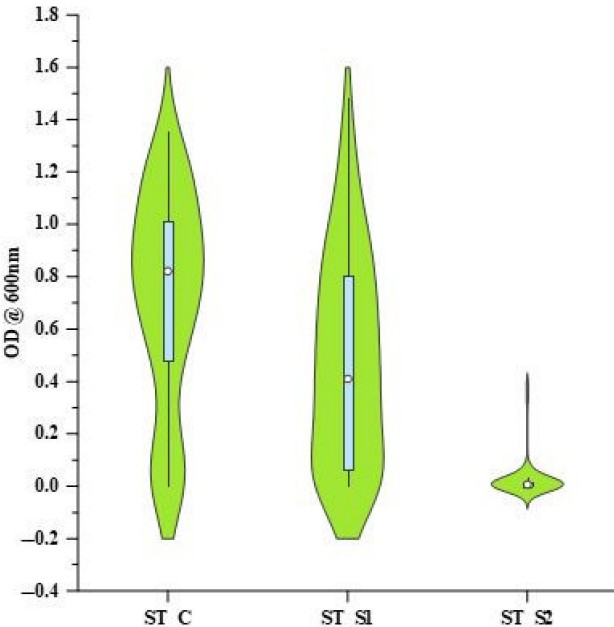

ST_C- Sodic tolerance Control, ST_1- Sodic tolerance @ 0.5 % Na$_2$CO$_3$ and ST_2 - Sodic tolerance @ 0.75 % Na$_2$CO$_3$

**Figure 3.** Bacterial growth in stressed and unstressed conditions. ST_C—Sodic tolerance for control, ST_S1—sodic tolerance at 0.5% Na$_2$CO$_3$ and ST_S2—sodic tolerance at 0.75% Na$_2$CO$_3$.

PCA was performed to choose the better EPS-producing STB isolates (Figure 4). The biplot showed that EPS production and bacterial growth under stress conditions were more influenced by parameters. The bacterial isolates were orthogonally spread over the four quadrants. The cultures laid in the (−, −) quadrant were considered to have less sodic tolerance with EPS production. Thus, the cultures spread across the (+, +), (+, −) and (−, +) quadrants (a total of 28 cultures) were chosen to examine their PGP activities (Figure 4).

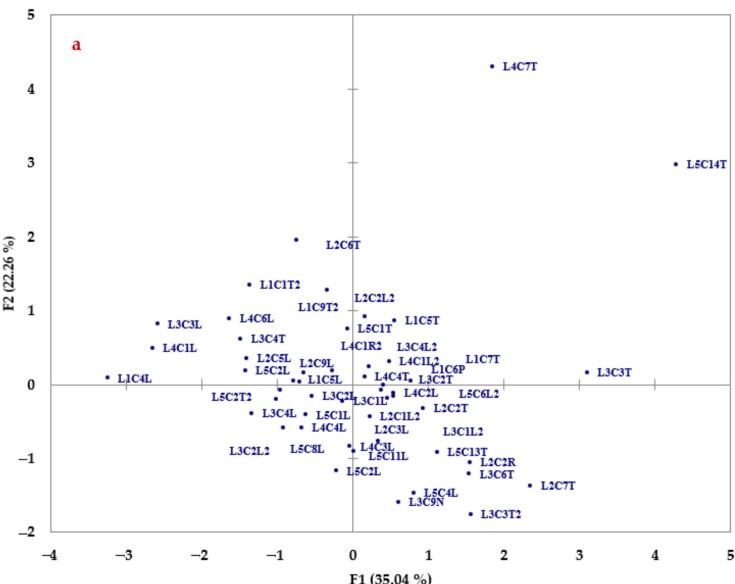

**Figure 4.** *Cont.*

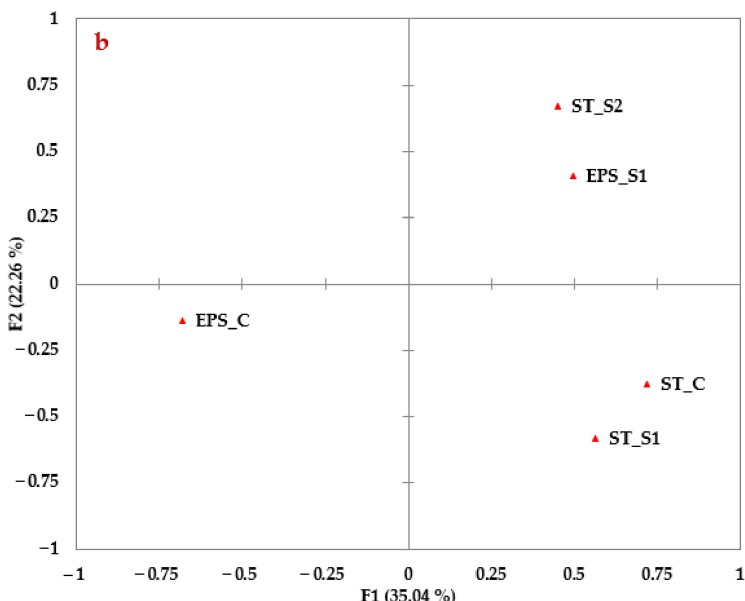

**Figure 4.** Principal component analysis plots relating EPS production, sodic tolerance, and bacterial isolates. (**a**) Biplot showing the position of active observation (bacterial isolates). (**b**) Biplot showing the position of active variables on F1 and F2. ST_C—sodic tolerance for control, ST_S1—sodic tolerance @ 0.5% $Na_2CO_3$, ST_S2—sodic tolerance @ 0.75% $Na_2CO_3$, EPS_C—EPS production @ Control, EPS_S1—EPS production @ 0.5% $Na_2CO_3$.

### 3.6. Assessing the Plant Growth Promoting Traits and Osmoprotectant Accumulation of Selected EPS-Producing Sodic-Tolerant Bacteria

PGP traits were examined for a selected 28 bacterial isolates. Under control conditions, *Cytobacillus firmus* L1C4L (8.8 µg/mL) followed by *Pseudomonas* sp. L5C14T (3.7 µg/mL) demonstrated higher IAA production, whereas under sodic stress conditions, the maximum production was noted in *Rhodococcus pyridinivorans* L3C9N and *Stenotrophomonas maltophilia* L3C7T. Sulphur oxidation was not detected in any of the tested bacterial isolates. The isolates *Bacillus safensis* L5C13T (120%), *Pseudomonas* sp. L5C14T (110%), and *Bacillus altitudinis* L3C3T2 (200%) solubilized the silica under unstressed conditions, and higher zinc and silica solubilization indices (600%, and 200%, respectively) were observed in *B. altitudinis* L3C3T2. Most bacterial isolates could not solubilize the phosphate, except *B. altitudinis* L3C3T2 (320%) and *Ralstonia picketti* L4C6L (200%). Siderophore production was found to be positive for *R. pyridinivorans* L3C9N, *Bacillus velezensis* L2C3L, *B. paralicheniformis* L1C5L, *Niallia circulans* L2C9L, *Burkholderia territorii* L2C6T, *Stenotrophomonas maltophilia* L2C7T, *Bacillus stercoris* L3C2T, *Bacillus cabrialesii* L4C3L, *Pseudomonas* sp. L5C14T, *Burkhold1eria* sp. L4C7T, *B. rugosus* L1C7T, *Bacillus tequilensis* L3C6T and *Bacillus coreaensis* L1C1T2 (Table 1). However, we could not create the sodicity conditions to find the nutrient solubilization capability of bacterial isolates in an agar medium. All the tested EPS-producing sodic-tolerant bacterial isolates produced proline, trehalose, and glycine betaine in unstress conditions. Stress caused a significant reduction in osmolyte accumulation compared to unstressed conditions. *B. safensis* L5C13T produced more proline under unstressed (53.7 mg/L) and stressed (12.8 mg/L) conditions. *B. cabrialesii* L1C5T yielded 226 mg/L of trehalose under stress-free conditions, whereas *B. altitudinis* L3C3T2 produced 149 mg/L of trehalose under stressed conditions. *B. cabrialesii* L1C5T (444 mg/L) and *Bacillus cereus* (275 mg/L) both produced higher glycine betaine in stress-free and stressful circumstances, respectively (Table 2).

**Table 1.** Plant growth promotion traits of screened EPS-producing sodic-tolerant bacteria isolated from sodic soils.

| Bacteria | IAA (mg/L) | | Zn Solubilization (%) | Siderophore Production |
|---|---|---|---|---|
| | **Without Stress** | **With Stress** | | |
| *Rhodococcus pyridinivorans* L3C9N | 2.5 ± 0.3 [ef] | 2.1 ± 0.4 [a] | – | + |
| *Bacillus velezensis* L2C3L | 1.4 ± 0.2 [i] | 0.6 ± 0.1 [hi] | – | + |
| *Cytobacillus firmus* L1C4L | 8.8 ± 0.3 [a] | 1.2 ± 0.1 [cd] | – | – |
| *Bacillus paralicheniformis* L1C5L | 2.9 ± 0.2 [cd] | 0.3 ± 0.1 [jk] | 260 | + |
| *Ralstonia pickettii* L4C6L | 0.7 ± 0.0 [jk] | 0.9 ± 0.2 [fg] | 520 | – |
| *Arthrobacter* sp. L5C8L | 2.6 ± 0.3 [efg] | 0.0 ± 0.0 [l] | – | – |
| *Niallia circulans* L2C9L | 1.9 ± 0.0 [hi] | 0.5 ± 0.1 [i] | – | + |
| *Staphylococcus* sp. L2C2T | 0.7 ± 0.0 [jk] | 0.1 ± 0.0 [kl] | – | – |
| *Burkholderia territorii* L2C6T | 0.7 ± 0.0 [jk] | 0.4 ± 0.1 [ij] | 188 | + |
| *Stenotrophomonas maltophilia* L2C7T | 1.9 ± 0.1 [hi] | 1.4 ± 0.1 [b] | – | + |
| *Bacillus stercoris* L3C2T | 1.5 ± 0.2 [hi] | 1.3 ± 0.2 [bc] | – | + |
| *Bacillus* sp. L3C3T | 0.9 ± 0.1 [j] | 1.1 ± 0.1 [cd] | – | – |
| *Bacillus cabrialesii* L1C5T | 0.5 ± 0.1 [k] | 0.8 ± 0.1 [gh] | – | + |
| *Bacillus safensis* L5C13T | 0.7 ± 0.0 [jk] | 0.8 ± 0.1 [gh] | 500 | – |
| *Pseudomonas* sp. L5C14T | 3.7 ± 0.1 [b] | 0.9 ± 0.0 [fg] | 186 | + |
| *Burkholderia cepacian* L4C7T | 2.2 ± 0.1 [g] | 1.0 ± 0.1 [de] | 300 | + |
| *Bacillus rugosus* L1C7T | 3.0 ± 0.2 [cd] | 0.6 ± 0.1 [i] | 140 | + |
| *Bacillus tequilensis* L3C6T | 1.7 ± 0.1 [hi] | 1.0 ± 0.2 [de] | – | + |
| *Kocuria* sp. L2C2R | 3.1 ± 0.2 [c] | 0.2 ± 0.0 [jk] | – | – |
| *Franconibacter helveticus* L2C1L2 | 0.9 ± 0.1 [j] | 1.1 ± 0.1 [cd] | – | – |
| *Bacillus paramycoides* L4C1L2 | 2.7 ± 0.3 [de] | 0.6 ± 0.1 [hi] | – | – |
| *Bacillus zanthoxyli* L2C2L2 | 2.6 ± 0.2 [ef] | 1.2 ± 0.0 [cd] | – | – |
| *Acinetobacter* sp. L5C6L2 | 2.3 ± 0.0 [fg] | 0.4 ± 0.1 [ij] | – | – |
| *Bacillus coreaensis* L1C1T2 | 2.5 ± 0.3 [efg] | 0.6 ± 0.0 [hi] | – | + |
| *Achromobacter* sp. L1C9T2 | 0.0 ± 0.0 [l] | 0.9 ± 0.0 [ef] | 300 | – |
| *Bacillus altitudinis* L3C3T2 | 2.3 ± 0.0 [g] | 1.0 ± 0.0 [de] | 600 | + |

Values in each column are the mean of three replications. Tukey's test was calculated at $p < 0.05$ to find the significance of the treatments. ± followed by numbers are standard deviation. (+)—positive production; (−)—negative production, IAA—indole acetic acid, Zn—zinc. Means sharing the same letters in each column are not significantly different as determined by the Tukey test ($p < 0.05$).

**Table 2.** Osmolyte production activities of screened EPS-producing sodic-tolerant bacteria isolated from sodic soils.

| Bacteria | Proline (mg/L) | | Trehalose (mg/L) | | Glycine Betaine (mg/L) | |
|---|---|---|---|---|---|---|
| | **Without Stress** | **With Stress** | **Without Stress** | **With Stress** | **Without Stress** | **With Stress** |
| *Rhodococcus pyridinivorans* L3C9N | 39.8 ± 5.1 [bc] | 8.2 ± 1.3 [cd] | 161 ± 22 [b] | 92 ± 15 [d] | 232 ± 34 [de] | 113 ± 01 [k] |
| *Bacillus velezensis* L2C3L | 16.4 ± 1.9 [ij] | 7.2 ± 0.8 [de] | 106 ± 16 [ef] | 52 ± 09 [fg] | 248 ± 39 [cd] | 124 ± 30 [jk] |
| *Cytobacillus firmus* L1C4L | 39.6 ± 1.7 [bc] | 4.8 ± 0.6 [ij] | 131 ± 13 [bc] | 16 ± 01 [lm] | 209 ± 24 [de] | 163 ± 11 [hi] |
| *Bacillus paralicheniformis* L1C5L | 29.2 ± 7.1 [ef] | 9.8 ± 1.9 [bc] | 156 ± 30 [bc] | 111 ± 23 [c] | 307 ± 58 [bc] | 139 ± 26 [ij] |
| *Ralstonia pickettii* L4C6L | 35.8 ± 7.1 [cd] | 5.5 ± 1.0 [fg] | 156 ± 40 [bc] | 14 ± 3.5 [lm] | 266 ± 66 [cd] | 155 ± 33 [hi] |
| *Arthrobacter* sp. L5C8L | 37.8 ± 2.0 [bc] | 7.4 ± 0.6 [de] | 75 ± 04 [hi] | 14 ± 01 [lm] | 263 ± 31 [cd] | 273 ± 50 [b] |

**Table 2.** *Cont.*

| Bacteria | Proline (mg/L) | | Trehalose (mg/L) | | Glycine Betaine (mg/L) | |
|---|---|---|---|---|---|---|
| | Without Stress | With Stress | Without Stress | With Stress | Without Stress | With Stress |
| *Niallia circulans* L2C9L | 43.8 ± 7.3 [bc] | 7.0 ± 1.1 [de] | 88 ± 15 [gh] | 42 ± 05 [hi] | 190 ± 18 [ef] | 374 ± 40 [a] |
| *Staphylococcus* sp. L2C2T | 47.4 ± 0.3 [ab] | 7.8 ± 0.3 [de] | 107 ± 04 [ef] | 65 ± 02 [ef] | 256 ± 07 [cd] | 225 ± 08 [cd] |
| *Burkholderia territorii* L2C6T | 40.1 ± 5.6 [bc] | 5.7 ± 0.7 [fg] | 122 ± 19 [de] | 52 ± 09 [fg] | 280 ± 50 [bcd] | 229 ± 26 [cd] |
| *Stenotrophomonas maltophilia* L2C7T | 47.3 ± 3.2 [ab] | 6.4 ± 0.3 [de] | 65 ± 06 [kl] | 51 ± 03 [fg] | 238 ± 17 [cd] | 226 ± 19 [cd] |
| *Bacillus stercoris* L3C2T | 38.1 ± 6.3 [bc] | 6.7 ± 1.0 [de] | 73 ± 11 [jk] | 61 ± 09 [ef] | 337 ± 15 [b] | 214 ± 01 [ef] |
| *Bacillus* sp. L3C3T | 46.4 ± 5.0 [ab] | 7.1 ± 0.6 [de] | 121 ± 11 [de] | 60 ± 03 [fg] | 254 ± 42 [cd] | 276 ± 42 [b] |
| *Bacillus cabrialesii* L1C5T | 46.5 ± 8.4 [ab] | 5.1 ± 8.4 [hi] | 225 ± 08 [a] | 35 ± 03 [jk] | 277 ± 50 [bc] | 262 ± 22 [bc] |
| *Bacillus safensis* L5C13T | 53.7 ± 7.7 [a] | 12.8 ± 7.7 [a] | 100 ± 08 [ef] | 26 ± 03 [kl] | 445 ± 77 [a] | 222 ± 33 [cd] |
| *Pseudomonas* sp. L5C14T | 35.5 ± 1.3 [cd] | 5.2 ± 1.3 [hi] | 56 ± 01 [l] | 41 ± 02 [ij] | 229 ± 06 [de] | 131 ± 01 [jk] |
| *Burkholderia cepacian* L4C7T | 29.3 ± 3.1 [ef] | 4.0 ± 3.1 [jk] | 124 ± 04 [de] | 55 ± 01 [g] | 234 ± 20 [de] | 111 ± 08 [k] |
| *Bacillus rugosus* L1C7T | 32.6 ± 7.7 [de] | 5.4 ± 7.7 [gh] | 72 ± 08 [jk] | 41 ± 01 [ij] | 168 ± 39 [gh] | 172 ± 40 [gh] |
| *Bacillus tequilensis* L3C6T | 35.9 ± 6.8 [cd] | 13.4 ± 6.8 [a] | 126 ± 07 [cd] | 50 ± 15 [gh] | 161 ± 43 [hi] | 142 ± 35 [ij] |
| *Kocuria* sp. L2C2R | 31.8 ± 3.4 [de] | 10.3 ± 3.4 [b] | 62 ± 03 [kl] | 59 ± 01 [fg] | 232 ± 12 [de] | 159 ± 12 [hi] |
| *Franconibacter helveticus* L2C1L2 | 35.0 ± 5.6 [cd] | 3.4 ± 5.9 [k] | 65 ± 06 [l] | 32 ± 02 [jk] | 128 ± 15 [i] | 164 ± 25 [hi] |
| *Bacillus paramycoides* L4C1L2 | 26.6 ± 2.7 [gh] | 3.3 ± 2.7 [k] | 62 ± 03 [kl] | 132.8 ± 03 [b] | 261 ± 25 [cd] | 210 ± 15 [ef] |
| *Bacillus zanthoxyli* L2C2L2 | 10.5 ± 0.2 [j] | 5.8 ± 0.2 [fg] | 89 ± 0.2 [gh] | 91.1 ± 02 [d] | 181 ± 05 [fg] | 151 ± 2.9 [ij] |
| *Acinetobacter* sp. L5C6L2 | 23.8 ± 4.2 [hi] | 5.9 ± 4.2 [ef] | 102 ± 04 [ef] | 74.1 ± 02 [ef] | 203 ± 34 [ef] | 127 ± 18 [jk] |
| *Bacillus coreaensis* L1C1T2 | 28.8 ± 1.2 [fg] | 1.3 ± 1.2 [l] | 90 ± 01 [fg] | 33.8 ± 01 [jk] | 165 ± 10 [gh] | 180 ± 09 [fg] |
| *Achromobacter* sp. L1C9T2 | 40.6 ± 1.7 [bc] | 3.4 ± 1.7 [k] | 140 ± 02 [bc] | 56.7 ± 02 [fg] | 190 ± 06 [ef] | 200 ± 01 [ef] |
| *Bacillus altitudinis* L3C3T2 | 36.3 ± 1.1 [cd] | 6.9 ± 0.1 [de] | 143 ± 02 [bc] | 149.1 ± 02 [a] | 241 ± 42 [cd] | 213 ± 30 [ef] |

Values in each column are the mean of three replications. Tukey's test was calculated at $p < 0.05$ to find the significance of the treatments. ± followed by numbers are standard deviation. Means sharing the same letters in each column are not significantly different as determined by the Tukey test ($p < 0.05$).

### 3.7. Screening the Biofilm-Forming Ability of Bacterial Isolates

PCA was carried out to select the potential biofilm-forming bacteria under sodic conditions (Figure 5 and Figure S4). Thus, biplots were created for biofilm formation (A550 nm) and biofilm/planktonic ratio (A550/A660 nm) for sodic (pH 9.5) and nonsodic conditions (control) at 4 and 8 days of incubation. Among the eight generated principal components, PC1 and PC2 had an eigenvalue of more than one and showed the total cumulative variations were 70.01%. All the weighted variables were spread over the (+, +) and (+, −) quadrants. The variables such as Biofim_S (biofilm formation under sodic stress), Biofilm/planktonic ratio_S (biofilm/planktonic ratio under sodic stress), Biofim_C (biofilm formation under unstressed conditions), and Biofilm/planktonic ratio_C (biofilm/planktonic ratio under unstressed conditions) observed on day 4 were the most deciding factor to find the potential biofilm-forming bacteria under sodic conditions. On consisting of these two plots, all the bacterial isolates were positioned over all four quadrants. The bacterial isolates placed on the (+, +) quadrant were high biofilm formers, which showed higher biofilm-forming ability for both 4 and 8 days of incubation, whereas those located on the (−, +) and (+, −) quadrants were moderate biofilm formers. The bacterial isolates positioned on the (−, −) quandrant had low biofilm-forming ability under sodic conditions due to their negative relationship with all variables. *Bacillus paralicheniformis* L1C5L, *Franconibacter helviticus* L2C1L2, *Pseudomonas* sp. L5C14T, *Arthrobacter* sp. L5C8L, and *Ralstonia pickettii* L4C6L showed strong biofilm-forming capability compared to the other isolates.

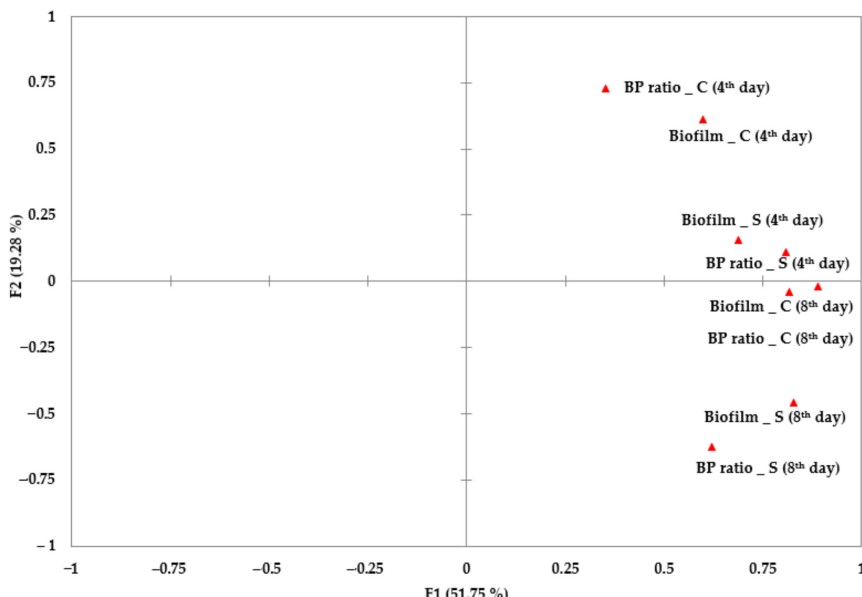

**Figure 5.** Principal component analysis plots relating biofilm-forming ability and EPS-producing sodic-tolerant bacterial isolates. Biofilm_C—biofilm production under control condition, Biofilm_S—biofilm production under sodic stress conditions, BP ratio_C—the ratio between biofilm production and planktonic population under control conditions, BP ratio_S—the ratio between biofilm production and planktonic population under stress conditions.

### 3.8. Intracellular Sodium Accumulation in Bacterial Cells

All five selected cultures did not accumulate sodium ions in their body.

### 3.9. Aggregation Dispersity Test

The soil particles were stabilized by water, and the supernatant was dispersed immediately with the addition of water. However, under treatment with the supernatant, the formed aggregates showed lesser dispersion and were relatively stronger than the control (Figure S2).

### 3.10. Alleviation of Sodic Stress

Distinct rice cultivars had different susceptibility ranges. As a result, establishing the ideal range of susceptibility for a particular variety becomes important. Figure 6 depicts the effect of sodium on rice seed germination. Root development slowed at 1.0 M sodium concentration. Only a few seeds germinated at 2.0 M, while the rest showed only shoot growth. In the concentration of 1.8 M, 40 percent seed germination was observed. As a result, the CO-51 cultivar's susceptibility range was set at 1.8 M. However, at this concentration, the plant's growth was hampered (data are not shown). Thus, the range at which root growth was affected (1.0 M sodium) was chosen for this study.

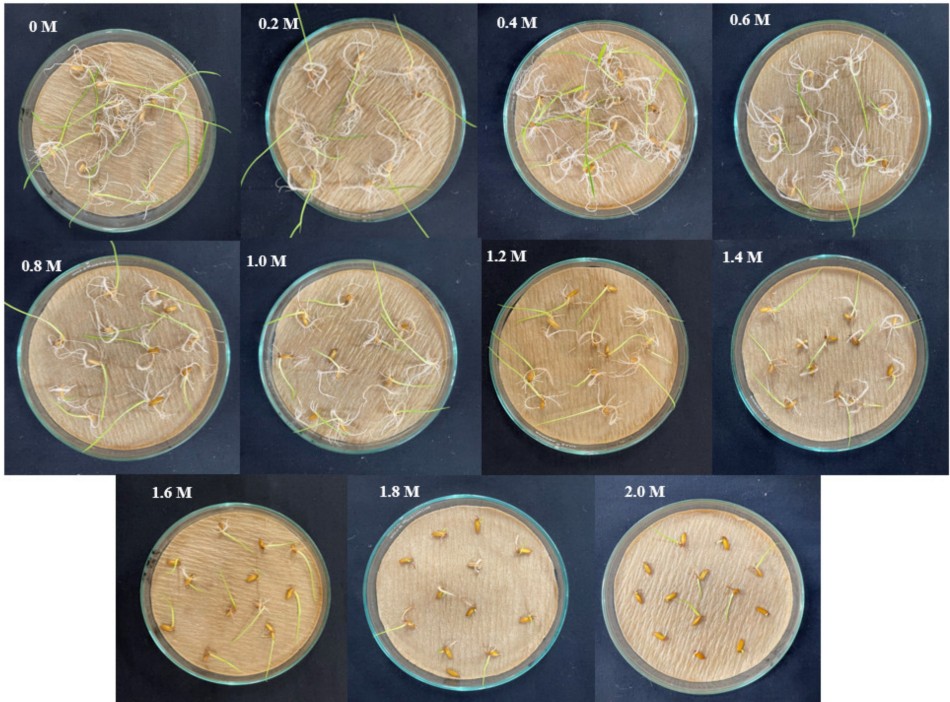

**Figure 6.** Optimization of sodic stress induction in selected cultivars. The concentration indicates the mole of sodium.

PGP activities were found more in unstressed conditions than under sodic stress, whereas the biofilm-forming ability was noticed well in both stressed and unstressed conditions. Thus, isolates were selected based on their sodic tolerance, EPS production, PGP activities, and biofilm-forming ability to validate the sodic stress alleviation in rice under sodic conditions. Hence, the well-performed isolates viz., *B. paralicheniformis* L1C5L, *Pseudomonas* sp. L5C14T, *B. rugosus* L1C7T, and *F. helveticus* L2C1L2 were chosen. The rice seeds were treated with the selected bacterial isolates and grown for 16 days under stress conditions (Figure S3). The un-inoculated seeds were kept as a control. After 16 days, the plant roots, shoot lengths, and dry weight were measured (Figure 7). The inoculation of *F. helveticus* L2C1L2 significantly enhanced root, shoot growth, and dry matter production more under stressed conditions. In addition, the inoculation of treatments significantly reduced $Na^+$ uptake per plant under sodic stress conditions. Inoculation of EPS-producing STB in rice plants reduced sodium uptake by 5.32% to 24.15% (Figure 7).

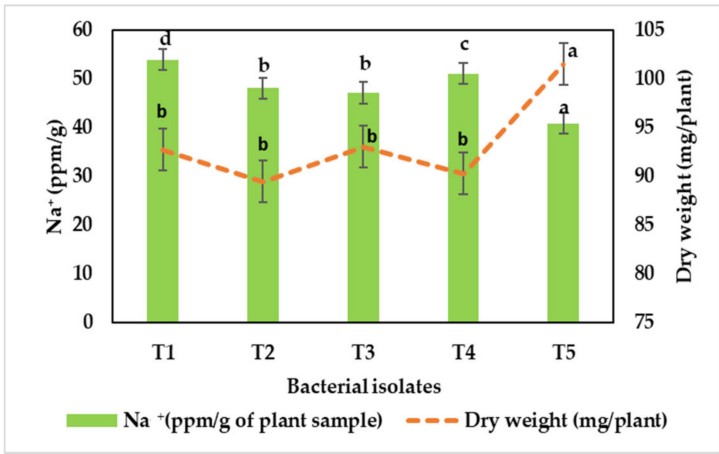

**Figure 7.** *Cont.*

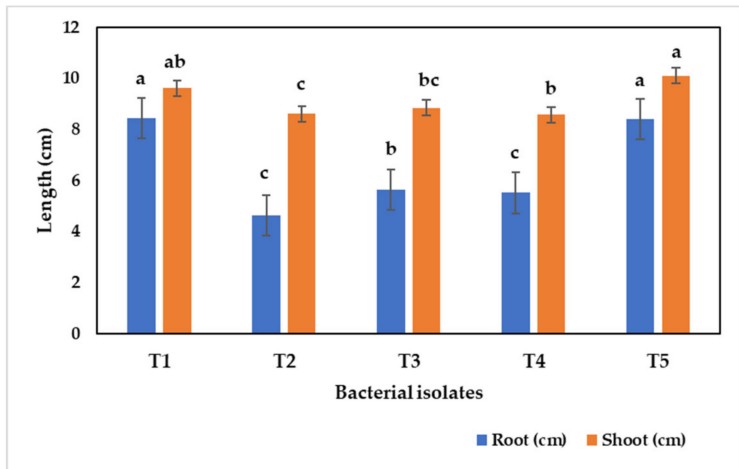

**Figure 7.** Impact of EPS-producing sodic-tolerant bacterial inoculation in rice on the plant growth and total dry weight and sodium absorption under sodic stress conditions. Each panel represent the mean of five replicates and the error bar indicates the standard error. The panels with the same letter are not significantly different as determined by the Tukey test ($p < 0.05$). T1—absolute control; T2—*Bacillus rugosus* L1C7T; T3—*Bacillus paralicheniformis* L1C5L; T4—*Pseudomonas* sp. L5C14T; T5—*Franconibacter helveticus* L2C1T2.

### 3.11. In Vitro Study on Biofilm Formation

An in vitro study confirmed the biofilm formation ability of *F. helveticus* L2C1T2 (Figure 8).

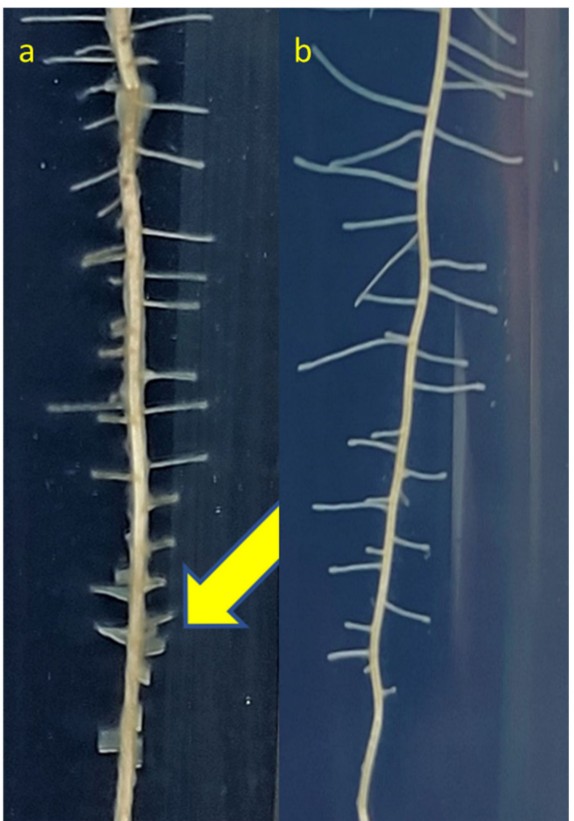

**Figure 8.** Biofilm formation on roots by *Franconibacter helveticus* L2C1L2 (**a**); control (**b**). The yellow arrow indicated the biofilm formation around the root by *F. helveticus* L2C1L2.

### 3.12. SEM Image

The morphological changes were seen in the SEM image of *F. helveticus* L2C1L2 under stressed and unstressed conditions (Figure 9). When compared to unstressed conditions, the length of the bacteria doubled, while the width was reduced by half under stressed conditions.

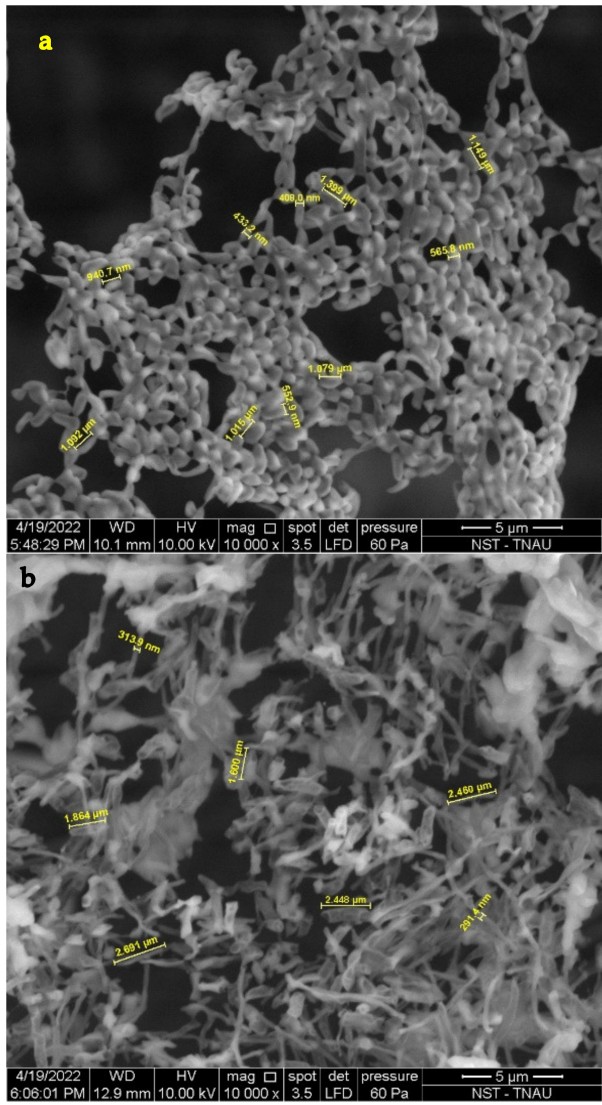

**Figure 9.** *Franconibacter helveticus* L2C1L2 under (**a**) unstressed and (**b**) stressed conditions.

### 4. Discussion

Sodicity (pH >8.5, EC < 4 dSm$^{-1}$ and ESP>15) is one of the salt stresses like saline (pH <8.5, EC > 4 dSm$^{-1}$ and ESP<15) and saline-sodic (pH < 8.5, EC > 4 dSm$^{-1}$ and ESP >15) and they are differentiated based on their pH, EC, and ESP [28]. Sodic stress impairs plant growth by creating specific ion toxicity and nutrient imbalances. To improve plant growth under sodic stress conditions, salt-tolerant beneficial microorganisms have been used for a long while. However, root colonization is also essential for microbes to establish a strong relationship with plants to improve plant growth and salt tolerance. Thus, utilization of EPS-producing STB-PGP bacteria can colonize and produce biofilm around the roots and enhance plant growth. The presence of EPS, as a major structural component in the biofilm, not only protects the microbes from the adverse environment but also binds the sodium ions that exist in the surrounding environment and reduces the uptake of sodium ions by plants. The present study showed that EPS-producing STB can be explored for plant growth promotion under sodic stress.

### 4.1. Media Optimization for Isolation of the Sodic Tolerant Bacteria

Diversified rice rhizosphere bacteria in sodic soil were isolated and screened for their EPS-producing ability. The EPS matrix produced by bacteria increases their survivability by protecting them from adverse conditions. In addition, it can retain water, create adhesion, communication, and aggregate with other cells. The highest EPS producer was chosen among the 253 isolated bacteria that were screened further for sodic tolerance and EPS production under sodic stress conditions to select the most efficient isolate which can survive under a natural sodic environment. Hence, it is more important to induce sodic stress in a growth medium to screen the bacterial isolates. Adequate screening techniques are essential and were varied according to the environment. For example, to choose the salt-tolerant bacteria in saline soil, NaCl was used, whereas in heavy metal-contaminated soil and alkaline soil, different concentrations of that heavy metal and $Na_2CO_3$ were used, respectively. Even though sodic soils differ from saline soil by having more sodium ions in their exchangeable sites and having a pH> 8.5, in numerous studies, NaCl was used to screen the sodic tolerant bacteria. Besides, sodium is present in the form of carbonate and bicarbonate in the sodic soil and during hydrolysis of $Na_2CO_3$ and $NaHCO_3$ release sodium ions and hydroxyl ions into the surrounding environment. This chemical reaction enhanced the sodium content and the pH. In this context, the addition of NaCl does not create appropriate sodic conditions in the medium since it is a neutral salt unable to produce pH > 7.5, but concomitantly, it yields more electrical conductivity. Thus, NaOH and $Na_2CO_3$ were chosen to induce the sodicity in the media. Though the inclusion of NaOH and $Na_2CO_3$ performed well in inducing sodicity, to mimic the sodic stress condition of soils, we used $Na_2CO_3$. Hence, we presumed that screening bacteria based on their tolerance to NaCl may affect their survivability in sodic soil under field conditions. An earlier study [4] reported that *Bacillus cereus* Pb25 under 9 $dSm^{-1}$ produced 19 mg/L of IAA, but in the present study *Bacillus* sp. L3C1L2 produced 1.13 mg/L under 0.5% $Na_2CO_3$ at pH 9.5. The *B. paralicheniformis* TRQ65 was isolated from salt-affected soils and could tolerate salt stress of 5% NaCl [29]. However, in the present study, the bacterial isolates could withstand up to 0.5% sodium carbonate at pH 9.5. Hence, the results from the present investigation revealed that the higher pH with higher specific ion concentration exhibited more impact on bacterial growth compared to the salinity alone. This was in line with the result suggested by Rousk et al. [30], who found that salinity was not the decisive factor in determining bacterial growth, but the other parameters, such as organic matter and pH, had more of an influence. Therefore, EPS production under sodic stress conditions and their sodic-tolerating ability were screened with 0.5% sodium carbonate and 9.5 pH. In total, 28 bacterial isolates were selected based on their EPS production and sodic tolerance. In this study, it was found that *Bacillus* spp. were more predominant in these sodic soils, which reflected the adverse conditions. Beyond their salt-tolerating ability, their spore-forming nature helps them to dominate over the other bacteria to survive in this adverse environment [11].

### 4.2. Influence of Sodic Stress on PGPtraits

The bacterial isolates exhibited more PGP traits in normal conditions than the stressed conditions. This has been reported by Soleimani et al. [31], and the same trend was observed in the present investigation. The ability of nutrient solubilisation and siderophore production under sodic stress conditions were unable to be assessed in laboratory conditions. It seemed too difficult to induce the sodic stress in the estimations except for IAA. IAA is an important PGP attribute produced by the bacteria isolated from the rhizosphere and is considered the most reliable method for selecting efficient PGPR [8]. Besides, in this study, the IAA production was reduced under stress conditions. Thus, extreme environments, such as sodic soil, and plant growth promotion not only depend on IAA production but also on osmolyte accumulation and root colonization [8]. Thus, the other properties, viz., biofilm formation and osmoprotectant production, were estimated. An enhanced biofilm-forming ability was noted in *B. paralicheniformis* L1C5L, *R. picketti* L4C6L,

*Pseudomonas* sp. L5C14T, *B. rugosus* L1C7T, *Burkholderia territorii* L2C6T, *S. maltophilia* L2C7T, *Kocuria* sp. L2C2R, *F. helveticus* L2C1L2, and *Acinetobacter* sp. L5C6L2 under sodic stress conditions. The biofilm-forming sodic-tolerant *R. picketti*, isolated from the rice rhizosphere, was identified previously from the pomegranate rhizosphere as an effective phosphate solubilizer [32] that was found in many clinical samples. *B. territorii* is found in soil, plant and human respiratory samples. Similarly, *S. maltophilia*, *Kocuria* sp., and *Acinetobacter* sp. are opportunistic human pathogens [33]. This result was supported by Balasundararajan and Dananjeyan [27], who found that the elite biofilm-forming bacteria in the plant rhizosphere could have both PGP characteristics and pathogenicity. In the present study, *B. paralicheniformis* L1C5L, *B. rugosus* L1C7T, and *F. helveticus* L2C1L2 have not been previously reported as a pathogen. The strong biofilm-forming ability of *B. paralicheniformis* on non-living material and its ability to thrive in a pH of 6 to 11 were reported earlier [34–37]. This might be the reason for the enhanced biofilm formation by *B. paralicheniformis* L1C5L seen in the current study with increasing stress and time course of incubation. Moreover, the presence of YmcA, Ylbf, and SinR as a regulator gene for biofilm formation in *B. paralicheniformis* regulates them to swarm, adhere and aggregate to form a complex [38,39]. For the first time, *F. helveticus* L2C1L2 was reported from the rice rhizosphere region under sodic soils to have EPS-producing sodic-tolerant capability with good biofilm-forming ability. The halotolerant and halophilic microorganisms adopt either the 'compatible solute' strategy or 'salt-in' strategy to survive in saline environments [40]. The compatible osmolytes strategy was embraced by both moderate halotolerant and halophilic bacteria, whereas the salt-in strategy purely occurred in true halophiles [41,42]. In our study, the selected bacterial isolates for the bioassay could be considered halotolerant because of their positive response to osmolyte accumulation rather than the sodium accumulation in vacuoles. Thus, the halotolerant nature of selected isolates could be expected to perform well under a sodic environment.

*4.3. Alleviation of Sodic Stress under In Vitro Condition*

The bioassay study was carried out to find effective bacteria to improve rice growth under sodic stress conditions. Among the treatments, the inoculation of *F. helveticus* L2C1L2 enhanced plant growth compared to the other treatments. The in vitro biofilm formation study supported biofilm production around the root (Figure 8). The enhanced biofilm formation under sodic stress conditions might be due to the enhanced aggregation of cells under stressed conditions compared to unstressed conditions. Typical cell structural changes and higher aggregation were noticed under the stressed conditions in the SEM image, supporting their adaptive nature in adverse environments. This was consistent with the result reported by Shultana et al. [16]. In addition, inoculation of *F. helveticus* L2C1L2 reduced sodium ion uptake in rice by 24% compared to uninoculated rice plants. The utilization of EPS-STB-PGPR could bind the sodium ions from the surrounding environment and reduce sodium uptake by the plant [11,16,43]. However, the sodic stress alleviation study was carried out in the roll towel method with a Hoagland solution. This might be the reason for the enhanced passive uptake of sodium in plants noticed in this study compared to the plants grown in the soil medium. Besides, the aggregation stability was improved by the EPS acting as an adhesive between the soil particles even under sodic conditions. Thus, the application of EPS-STB enhanced the shoot and root growth under sodic conditions by reducing the sodium uptake and improving the soil aggregation stability.

## 5. Conclusions

From the present investigation, we concluded that the better EPS-producing STB are screened with sodium carbonate that is capable of surviving under sodic conditions. The inoculation of sodic-tolerant EPS-producing PGP bacterium (*F. helveticus* L2C1L2) in rice could significantly improve plant growth under stress conditions by producing a biofilm around the root and affecting the sodium ion uptake. In addition, the EPS production enhanced the aggregate formation in the dispersed sodic soil. Therefore, *F. helveticus*

L2C1L2 would be a suitable bacterium for lowering sodic stress in rice grown in sodic soil. However, a careful examination of virulence activities in selected isolates is needed, and co-inoculation of endophytes and EPS-producing sodic-tolerant biofilm-forming inoculants will become a good management practice to enhance systemic tolerance in plants for alleviating sodic stress.

**Supplementary Materials:** The following supporting information can be downloaded at: https://www.mdpi.com/article/10.3390/agriculture12091451/s1, Figure S1: Screening process for selecting the EPS-producing sodic-tolerant PGPR; Figure S2: Aggregate formation in dispersed soil; Figure S3: Impact of bacterial inoculation on plant growth under sodic stress and unstressed conditions; Figure S4: Principal component analysis plots relating biofilm-forming ability and EPS-producing sodic-tolerant bacterial isolates; Table S1: Soil properties of collected sodic soils; Table S2: Bacteria isolated from the rhizosphere of rice under sodic soils; Table S3: Selection of sodium sources.

**Author Contributions:** R.A. and S.T. designed the work and corrected the manuscript. Y.G. performed the work. M.A. helped in the analysis of the work. A.R.C. and J.-H.P. sequenced the isolated cultures. D.S. and L.C. took part in sample collections and analysis. Y.G. analyzed the data and drafted the manuscript. All authors have read and agreed to the published version of the manuscript.

**Funding:** This research received no external funding.

**Institutional Review Board Statement:** There was no human or animal subjects involved in this study.

**Data Availability Statement:** All sequence data generated in this study were deposited in NCBI GenBank (https://www.ncbi.nlm.nih.gov/, 21 December 2021) under the accession no. OK427230, OM392062-OM392064, OM421749, OM421787-OM421792, OM422610-OM422640, OM486948, OM584324, OM604753, OM614587, OM615901, OM614598, OM615903, OM616571, and OM640465.

**Conflicts of Interest:** The authors declare no conflict of interest.

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
