# Peer review of "Alleviation of Sodic Stress in Rice by Exploring the Exopolysaccharide-Producing Sodic-Tolerant Bacteria"

_agriculture, doi:10.3390/agriculture12091451_

Round 1

Reviewer 1 Report

The overall manuscript is written very well. However, before acceptance author should improve these mistakes.

Line 17, rewrite line “Chinna Salem, Tamil Nadu, and India”

Line 17 and 36, both lines similar structure, rewrite at least one place.

Line 38 to 41, provide suitable reference.

Line 44 to 46, provide suitable reference.

Line 50. Use full form at first use “PGPR”

Line 50-51 “Inoculation of PGPR is advocated to increase the growth of 50 rice, wheat, and sunflower under sodic stress conditions” provide suitable reference at the end of lines.

Line 56, remove typo “. [5, 6]. Remove full stop.

Line 99 to 100 “The plates were incubated for 48 h at 30ËšC and surveyed every 24 h for new bacterial colo- 100 nies” how you optimized incubated time and temperature?

Line 360 and 361, remove typo, heading separate from paragraph.

The decimal of value reported in all tables of the manuscript must be uniform as the value of the detection limit of the used method.

Line 390 Use full form at first use “PCA”

Please correct all grammatical errors and typos

Please carefully align the formatting of citations in main text, also recheck all the citations with references for avoiding any repetition or missing citation.

-The abbreviations are needed to follow the standard format at all places, also avoid repeating full form at second use and so on

Please add concluding lines in the abstract

Reviewer 2 Report

View letter

Manuscript No.: agriculture-1871051

Title: Characterization of exopolysaccharide producing sodic tolerant bacterial diversity under a rice-sodic environment to alleviate the sodic stress

The experimental design process of this paper is clear and the data are detailed, which can better support the research and development of salt-tolerant bacteria. However, there are still some problems to be discussed and improved. The specific suggestions are as follows:

1. The title is somewhat mouthful, for example, Sodic appears many times. So the title need to be more clear and highlight the theme.

2. Keyword: Rice-sodic soil

3. Line 50, what about the PGPR ?

4. Line 84, the rice plants were selected from the sodic soil. Which growth period ?

5. Line 85, do you measure the OH- or CO32- or HCO3- in the rice soil environment ? Which decide the pH ?

6. Line 109, maybe the NaCl and NaOH is better.

7. Line 136, is there have any references support the universal primer ? That is for sodic ?

8. Line 256, Na+ .

9. Line 281, The SEM image may not reflect the real features after dry and dehydration ?

10. Line 305, how to select the fifty bacterial ? are they huge or effective ? That is more importance for the following analysis.

11. Line 336, does CO32- influence the result ?

12. Figure 4A, form the PCA analysis, there is not more difference on the bacterial and the explanation rate is relatively low ?

13. Table 1, Is there have any relationship between IAA con. and siderophore production (+ or -). From the IAA, the sodic stress inhibit the IAA release for most bacterial.

14. Figure 7, lack of significance of the treatments.

15. The “Discussion” need at lest two subtitle for the biofilm characteristic and the effect of salt tolerance.

Reviewer 3 Report

Dear Authors

The area of saline soils will apparently increase due to modern climatic changes. The agricultural use of saline soils is severely limited. The search for microorganisms to mitigate sodium stress is of scientific and practical interest.

In general, I agree with the screening scheme for EPS-producing PGP bacteria that the authors implemented. The authors tested the bacteria for many plant growth promoting traits. But not all traits have been discussed in relation to sodium stress mitigation in plants. For example, production of IAA and siderophores, solubilization of phosphorus and zinc.

Research on sodium stress mitigation in rice seems to be incomplete. In addition to the physiological and biochemical properties of bacteria, the stimulating effect also depends on which substrate (solution, sand, soil, flooded soil) inside the roots and bacteria interact. It also depends on the interaction of introduced bacteria with native microorganisms. Usually, growing plants in rolls allows only preliminary conclusions to be drawn. Do you have other results of treating rice grown in non-sterile soil similar to the one from which you isolated the bacteria?

Comments

1)      The use of the term bacterial diversity in the title seems unfortunate to me. It is more suitable for studying bacterial communities, the number of species in communities, and the frequency of occurrence of species. For the authors bacterial communities of sodic soils  were only a source of isolates. The word "diversity" hardly appears in the text.

2)      The authors describe the screening of microorganisms, which included quite a few steps. It is not easy to keep track and remember how many strains were rejected at each stage and which strains passed to the next stage. Perhaps a diagram illustrating the screening process could make this information easier to understand.

3)      Research methods are described in detail and clearly. However, I need one clarification. Line 269. What was the cell-free extract? Is this the supernatant after centrifugation? Is it a polysaccharide extracted according to point 2.3?

4)      Some figures are missing or duplicated in the manuscript. For example, figures 2 and 3, part of figure 5 describing the isolates.

5)      Line 451. It is better to briefly describe the changes that are clearly visible to you. Readers may find that the most important differences are not the ones you imply.

6)      Figure 7. Due to the small size of the "roots" and "shoots", the difference between the treatment options is difficult to detect. I propose to divide the figure into two diagrams or add letters indicating a significant difference. The description of Figure 7 says «The inoculation of bacterial isolates enhanced root growth more under stressed conditions». It seems to me that all strains except Franconibacter helveticus L2C1T2 inhibited root growth.

7)      Line 459 – 461. Maybe instead of the word «improvise» you were going to use «improve»?

8)       Line 556-568. It seems to me that there are inaccuracies in the conclusions. «From the present investigation, we concluded that the better EPS producing sodic tolerant bacteria are screened with sodium carbonate that is capable of surviving under sodic conditions». In this study, you did not compare sodium chloride screening with sodium carbonate screening. Therefore, it is incorrect to write that you have shown the benefits of screening with sodium carbonate.  «The inoculation of sodic tolerant EPS producing plant growth promoting bacterium (Franconibacter helveticus) in rice significantly improved the plant growth under stress conditions by producing biofilm around the root and affecting the sodium ion uptake». Data on the absorption of sodium by rice plants are not available in the manuscript. This is your assumption, which requires experimental verification. It is better to write it in the form of an assumption, not a statement.

Round 2

Reviewer 1 Report

Aceepted.

Reviewer 2 Report

View letter

No.: agriculture-1871051

Title: Characterization of exopolysaccharide producing sodic tolerant bacterial diversity under a rice-sodic environment to alleviate the sodic stress

The author has made a good revision, which basically meets the requirements. However, I still have a question that for the trial, the soil pH > 8.0, which means that the soil was strongly alkaline. From the Method, 2.7 and 2.8, there use pH 7.0 and 9.5 which had more impact than Na+ con. on bacterial. The pH is one important factor on rice growth and bacterial life. For rice growth, the soil pH 6.5-7.5 is better. Therefore, the theme mainly producing saline-alkaline tolerance.

Reviewer 3 Report

A few comments

Line 254: model and manufacturer of the photometer?

Figure 3: ST_S1 or ST_1? ST_S2 or ST_2?

Figure 4 a: the x-axis signatures overlap
